# Operator Variational Inference

**Rajesh Ranganath**
Princeton University

**Jaan Altosaar**
Princeton University

**Dustin Tran**
Columbia University

**David M. Blei**
Columbia University

## Abstract

Variational inference is an umbrella term for algorithms which cast Bayesian inference as optimization. Classically, variational inference uses the Kullback-Leibler divergence to define the optimization. Though this divergence has been widely used, the resultant posterior approximation can suffer from undesirable statistical properties. To address this, we reexamine variational inference from its roots as an optimization problem. We use *operators*, or functions of functions, to design variational objectives. As one example, we design a variational objective with a Langevin-Stein operator. We develop a black box algorithm, operator variational inference (OPVI), for optimizing any operator objective. Importantly, operators enable us to make explicit the statistical and computational tradeoffs for variational inference. We can characterize different properties of variational objectives, such as objectives that admit *data subsampling*—allowing inference to scale to massive data—as well as objectives that admit *variational programs*—a rich class of posterior approximations that does not require a tractable density. We illustrate the benefits of OPVI on a mixture model and a generative model of images.

## 1 Introduction

Variational inference is an umbrella term for algorithms that cast Bayesian inference as optimization [10]. Originally developed in the 1990s, recent advances in variational inference have scaled Bayesian computation to massive data [7], provided black box strategies for generic inference in many models [19], and enabled more accurate approximations of a model's posterior without sacrificing efficiency [21, 20]. These innovations have both scaled Bayesian analysis and removed the analytic burdens that have traditionally taxed its practice.

Given a model of latent and observed variables $p(x, z)$, variational inference posits a family of distributions over its latent variables and then finds the member of that family closest to the posterior, $p(z \mid x)$. This is typically formalized as minimizing a Kullback-Leibler (KL) divergence from the approximating family $q(\cdot)$ to the posterior $p(\cdot)$. However, while the $\text{KL}(q \parallel p)$ objective offers many beneficial computational properties, it is ultimately designed for convenience; it sacrifices many desirable statistical properties of the resultant approximation.

When optimizing KL, there are two issues with the posterior approximation that we highlight. First, it typically underestimates the variance of the posterior. Second, it can result in degenerate solutions that zero out the probability of certain configurations of the latent variables. While both of these issues can be partially circumvented by using more expressive approximating families, they ultimately stem from the choice of the objective. Under the KL divergence, we pay a large price when $q(\cdot)$ is big where $p(\cdot)$ is tiny; this price becomes infinite when $q(\cdot)$ has larger support than $p(\cdot)$.

In this paper, we revisit variational inference from its core principle as an optimization problem. We use *operators*—mappings from functions to functions—to design variational objectives, explicitly trading off computational properties of the optimization with statistical properties of the approximation. We use operators to formalize the basic properties needed for variational inference algorithms. We further outline how to use them to define new variational objectives; as one example, we design a variational objective using a Langevin-Stein operator.

We develop operator variational inference (OPVI), a black box algorithm that optimizes any operator objective. In the context of OPVI, we show that the Langevin-Stein objective enjoys two good properties. First, it is amenable to *data subsampling*, which allows inference to scale to massive data. Second, it permits rich approximating families, called *variational programs*, which do not require analytically tractable densities. This greatly expands the class of variational families and the fidelity of the resulting approximation. (We note that the traditional KL is not amenable to using variational programs.) We study OPVI with the Langevin-Stein objective on a mixture model and a generative model of images.

**Related Work.** There are several threads of research in variational inference with alternative divergences. An early example is expectation propagation (EP) [16]. EP promises approximate minimization of the inclusive KL divergence $KL(p||q)$ to find overdispersed approximations to the posterior. EP hinges on local minimization with respect to subsets of data and connects to work on $\alpha$-divergence minimization [17, 6]. However, it does not have convergence guarantees and typically does not minimize KL or an $\alpha$-divergence because it is not a global optimization method. We note that these divergences can be written as operator variational objectives, but they do not satisfy the tractability criteria and thus require further approximations. Li and Turner [14] present a variant of $\alpha$-divergences that satisfy the full requirements of OPVI. Score matching [9], a method for estimating models by matching the score function of one distribution to another that can be sampled, also falls into the class of objectives we develop.

Here we show how to construct new objectives, including some not yet studied. We make explicit the requirements to construct objectives for variational inference. Finally, we discuss further properties that make them amenable to both scalable and flexible variational inference.

## 2 Operator Variational Objectives

We define operator variational objectives and the conditions needed for an objective to be useful for variational inference. We develop a new objective, the Langevin-Stein objective, and show how to place the classical KL into this class. In the next section, we develop a general algorithm for optimizing operator variational objectives.

### 2.1 Variational Objectives

Consider a probabilistic model $p(x, z)$ of data $x$ and latent variables $z$. Given a data set $x$, approximate Bayesian inference seeks to approximate the posterior distribution $p(z \mid x)$, which is applied in all downstream tasks. Variational inference posits a family of approximating distributions $q(z)$ and optimizes a divergence function to find the member of the family closest to the posterior.

The divergence function is the *variational objective*, a function of both the posterior and the approximating distribution. Useful variational objectives hinge on two properties: first, optimizing the function yields a good posterior approximation; second, the problem is tractable when the posterior distribution is known up to a constant.

The classic construction that satisfies these properties is the evidence lower bound (ELBO),

$$\mathbb{E}_{q(z)}[\log p(x, z) - \log q(z)]. \tag{1}$$

It is maximized when $q(z) = p(z \mid x)$ and it only depends on the posterior distribution up to a tractable constant, $\log p(x, z)$. The ELBO has been the focus in much of the classical literature. Maximizing the ELBO is equivalent to minimizing the KL divergence to the posterior, and the expectations are analytic for a large class of models [4].

### 2.2 Operator Variational Objectives

We define a new class of variational objectives, *operator variational objectives*. An operator objective has three components. The first component is an operator $O^{p,q}$ that depends on $p(z \mid x)$ and $q(z)$. (Recall that an operator maps functions to other functions.) The second component is a family of test functions $\mathcal{F}$, where each $f(z) \in \mathcal{F}$ maps realizations of the latent variables to real vectors $\mathbb{R}^d$. In the objective, the operator and a function will combine in an expectation $\mathbb{E}_{q(z)}[(O^{p,q} f)(z)]$, designed such that values close to zero indicate that $q$ is close to $p$. The third component is a distance

function $t(a) : \mathbb{R} \to [0, \infty)$, which is applied to the expectation so that the objective is non-negative. (Our example uses the square function $t(a) = a^2$.)

These three components combine to form the operator variational objective. It is a non-negative function of the variational distribution,

$$\mathcal{L}(q; O^{p,q}, \mathcal{F}, t) = \sup_{f \in \mathcal{F}} t(\mathbb{E}_{q(z)}[(O^{p,q} f)(z)]). \tag{2}$$

Intuitively, it is the worst-case expected value among all test functions $f \in \mathcal{F}$. Operator variational inference seeks to minimize this objective with respect to the variational family $q \in \mathcal{Q}$.

We use operator objectives for posterior inference. This requires two conditions on the operator and function family.

1. *Closeness*. The minimum of the variational objective is at the posterior, $q(z) = p(z \mid x)$. We meet this condition by requiring that $\mathbb{E}_{p(z \mid x)}[(O^{p,p} f)(z)] = 0$ for all $f \in \mathcal{F}$. Thus, optimizing the objective will produce $p(z \mid x)$ if it is the only member of $\mathcal{Q}$ with zero expectation (otherwise it will produce a distribution in the equivalence class: $q \in \mathcal{Q}$ with zero expectation). In practice, the minimum will be the closest member of $\mathcal{Q}$ to $p(z \mid x)$.

2. *Tractability*. We can calculate the variational objective up to a constant without involving the exact posterior $p(z \mid x)$. In other words, we do not require calculating the normalizing constant of the posterior, which is typically intractable. We meet this condition by requiring that the operator $O^{p,q}$—originally in terms of $p(z \mid x)$ and $q(z)$—can be written in terms of $p(x, z)$ and $q(z)$. Tractability also imposes conditions on $\mathcal{F}$: it must be feasible to find the supremum. Below, we satisfy this by defining a parametric family for $\mathcal{F}$ that is amenable to stochastic optimization.

Equation 2 and the two conditions provide a mechanism to design meaningful variational objectives for posterior inference. Operator variational objectives try to match expectations with respect to $q(z)$ to those with respect to $p(z \mid x)$.

## 2.3 Understanding Operator Variational Objectives

Consider operators where $\mathbb{E}_{q(z)}[(O^{p,q} f)(z)]$ only takes positive values. In this case, distance to zero can be measured with the identity $t(a) = a$, so tractability implies the operator need only be known up to a constant. This family includes tractable forms of familiar divergences like the KL divergence (ELBO), Rényi's $\alpha$-divergence [14], and the $\chi$-divergence [18].

When the expectation can take positive or negative values, operator variational objectives are closely related to Stein divergences [2]. Consider a family of scalar test functions $\mathcal{F}^*$ that have expectation zero with respect to the posterior, $\mathbb{E}_{p(z \mid x)}[f^*(z)] = 0$. Using this family, a *Stein divergence* is

$$D_{\text{Stein}}(p, q) = \sup_{f^* \in \mathcal{F}^*} |\mathbb{E}_{q(z)}[f^*(z)] - \mathbb{E}_{p(z \mid x)}[f^*(z)]|.$$

Now recall the operator objective of Equation 2. The closeness condition implies that

$$\mathcal{L}(q; O^{p,q}, \mathcal{F}, t) = \sup_{f \in \mathcal{F}} t(\mathbb{E}_{q(z)}[(O^{p,q} f)(z)] - \mathbb{E}_{p(z \mid x)}[(O^{p,p} f)(z)]).$$

In other words, operators with positive or negative expectations lead to Stein divergences with a more generalized notion of distance.

## 2.4 Langevin-Stein Operator Variational Objective

We developed the operator variational objective. It is a class of tractable objectives, each of which can be optimized to yield an approximation to the posterior. An operator variational objective is built from an operator, function class, and distance function to zero. We now use this construction to design a new type of variational objective.

An operator objective involves a class of functions that has known expectations with respect to an intractable distribution. There are many ways to construct such classes [1, 2]. Here, we construct an operator objective from the generator Stein's method applied to the Langevin diffusion.

Let $\nabla^{\top} f$ denote the divergence of a vector-valued function $f$, that is, the sum of its individual gradients. Applying the generator method of Barbour [2] to Langevin diffusion gives the operator

$$(O_{\text{LS}}^{p} f)(z) = \nabla_z \log p(x, z)^{\top} f(z) + \nabla^{\top} f. \tag{3}$$

We call this the Langevin-Stein (LS) operator. We obtain the corresponding variational objective by using the squared distance function and substituting Equation 3 into Equation 2,

$$\mathcal{L}(q; O_{\text{LS}}^{p}, \mathcal{F}) = \sup_{f \in \mathcal{F}} (\mathbb{E}_q [\nabla_z \log p(x, z)^{\top} f(z) + \nabla^{\top} f])^2. \tag{4}$$

The LS operator satisfies both conditions. First, it satisfies closeness because it has expectation zero under the posterior (Appendix A) and its unique minimizer is the posterior (Appendix B). Second, it is tractable because it requires only the joint distribution. The functions $f$ will also be a parametric family, which we detail later.

Additionally, while the KL divergence finds variational distributions that underestimate the variance, the LS objective does not suffer from that pathology. The reason is that KL is infinite when the support of $q$ is larger than $p$; here this is not the case.

We provided one example of a variational objectives using operators, which is specific to continuous variables. In general, operator objectives are not limited to continuous variables; Appendix C describes an operator for discrete variables.

## 2.5 The KL Divergence as an Operator Variational Objective

Finally, we demonstrate how classical variational methods fall inside the operator family. For example, traditional variational inference minimizes the KL divergence from an approximating family to the posterior [10]. This can be construed as an operator variational objective,

$$(O_{\text{KL}}^{p;q} f)(z) = \log q(z) - \log p(z|x) \quad \forall f \in \mathcal{F}. \tag{5}$$

This operator does not use the family of functions—it trivially maps all functions $f$ to the same function. Further, because KL is strictly positive, we use the identity distance $t(a) = a$.

The operator satisfies both conditions. It satisfies closeness because $\text{KL}(p||p) = 0$. It satisfies tractability because it can be computed up to a constant when used in the operator objective of Equation 2. Tractability comes from the fact that $\log p(z \,|\, x) = \log p(z, x) - \log p(x)$.

# 3 Operator Variational Inference

We described operator variational objectives, a broad class of objectives for variational inference. We now examine how it can be optimized. We develop a black box algorithm [27, 19] based on Monte Carlo estimation and stochastic optimization. Our algorithm applies to a general class of models and any operator objective.

Minimizing the operator objective involves two optimizations: minimizing the objective with respect to the approximating family $\mathcal{Q}$ and maximizing the objective with respect to the function class $\mathcal{F}$ (which is part of the objective).

We index the family $\mathcal{Q}$ with *variational parameters* $\lambda$ and require that it satisfies properties typically assumed by black box methods [19]: the variational distribution $q(z; \lambda)$ has a known and tractable density; we can sample from $q(z; \lambda)$; and we can tractably compute the score function $\nabla_\lambda \log q(z; \lambda)$. We index the function class $\mathcal{F}$ with parameters $\theta$, and require that $f_\theta(\cdot)$ is differentiable. In the experiments, we use neural networks, which are flexible enough to approximate a general family of test functions [8].

Given parameterizations of the variational family and test family, operator variational inference (OPVI) seeks to solve a minimax problem,

$$\lambda^* = \inf_{\lambda} \sup_{\theta} t(\mathbb{E}_\lambda [(O^{p,q} f_\theta)(z)]). \tag{6}$$

We will use stochastic optimization [23, 13]. In principle, we can find stochastic gradients of $\lambda$ by rewriting the objective in terms of the optimized value of $\theta$, $\theta^*(\lambda)$. In practice, however, we

---

**Algorithm 1:** Operator variational inference

---

**Input** : Model log $p(x, z)$, variational approximation $q(z; \lambda)$
**Output**: Variational parameters $\lambda$
Initialize $\lambda$ and $\theta$ randomly.
**while** *not converged* **do**
  Compute unbiased estimates of $\nabla_\lambda \mathcal{L}_\theta$ from Equation 7.
  Compute unbiased esimates of $\nabla_\theta \mathcal{L}_\lambda$ from Equation 8.
  Update $\lambda$, $\theta$ with unbiased stochastic gradients.
**end**

---

simultaneously solve the maximization and minimization. Though computationally beneficial, this produces saddle points. In our experiments we found it to be stable enough. We derive gradients for the variational parameters $\lambda$ and test function parameters $\theta$. (We fix the distance function to be the square $t(a) = a^2$; the identity $t(a) = a$ also readily applies.)

**Gradient with respect to $\lambda$.** For a fixed test function with parameters $\theta$, denote the objective

$$\mathcal{L}_\theta = t(\mathbb{E}_\lambda[(O^{p,q} \ f_\theta)(z)]).$$

The gradient with respect to variational parameters $\lambda$ is

$$\nabla_\lambda \mathcal{L}_\theta = 2 \, \mathbb{E}_\lambda[(O^{p,q} \ f_\theta)(z)] \, \nabla_\lambda \mathbb{E}_\lambda[(O^{p,q} \ f_\theta)(z)].$$

Now write the second expectation with the score function gradient [19]. This gradient is

$$\nabla_\lambda \mathcal{L}_\theta = 2 \, \mathbb{E}_\lambda[(O^{p,q} \ f_\theta)(z)] \, \mathbb{E}_\lambda[\nabla_\lambda \log q(z; \lambda)(O^{p,q} \ f_\theta)(z) + \nabla_\lambda(O^{p,q} \ f_\theta)(z)]. \quad (7)$$

Equation 7 lets us calculate unbiased stochastic gradients. We first generate two sets of independent samples from $q$; we then form Monte Carlo estimates of the first and second expectations. For the second expectation, we can use the variance reduction techniques developed for black box variational inference, such as Rao-Blackwellization [19].

We described the score gradient because it is general. An alternative is to use the reparameterization gradient for the second expectation [11, 22]. It requires that the operator be differentiable with respect to $z$ and that samples from $q$ can be drawn as a transformation $r$ of a parameter-free noise source $\epsilon$, $z = r(\epsilon, \lambda)$. In our experiments, we use the reparameterization gradient.

**Gradient with respect to $\theta$.** Mirroring the notation above, the operator objective for fixed variational $\lambda$ is

$$\mathcal{L}_\lambda = t(\mathbb{E}_\lambda[(O^{p,q} \ f_\theta)(z)]).$$

The gradient with respect to test function parameters $\theta$ is

$$\nabla_\theta \mathcal{L}_\lambda = 2 \, \mathbb{E}_\lambda[(O^{p,q} f_\theta)(z)] \, \mathbb{E}_\lambda[\nabla_\theta O^{p,q} \ f_\theta(z)]. \quad (8)$$

Again, we can construct unbiased stochastic gradients with two sets of Monte Carlo estimates. Note that gradients for the test function do not require score gradients (or reparameterization gradients) because the expectation does not depend on $\theta$.

**Algorithm.** Algorithm 1 outlines OPVI. We simultaneously minimize the variational objective with respect to the variational family $q_\lambda$ while maximizing it with respect to the function class $f_\theta$. Given a model, operator, and function class parameterization, we can use automatic differentiation to calculate the necessary gradients [3]. Provided the operator does not require model-specific computation, this algorithm satisfies the black box criteria.

### 3.1 Data Subsampling and OPVI

With stochastic optimization, data subsampling scales up traditional variational inference to massive data [7, 26]. The idea is to calculate noisy gradients by repeatedly subsampling from the data set, without needing to pass through the entire data set for each gradient.

An as illustration, consider hierarchical models. Hierarchical models consist of global latent variables $\beta$ that are shared across data points and local latent variables $z_i$ each of which is associated to a data point $x_i$. The model's log joint density is

$$\log p(x_{1:n}, z_{1:n}, \beta) = \log p(\beta) + \sum_{i=1}^{n} \Big[ \log p(x_i \mid z_i, \beta) + \log p(z_i \mid \beta) \Big].$$

Hoffman et al. [7] calculate unbiased estimates of the log joint density (and its gradient) by subsampling data and appropriately scaling the sum.

We can characterize whether OPVI with a particular operator supports data subsampling. OPVI relies on evaluating the operator and its gradient at different realizations of the latent variables (Equation 7 and Equation 8). Thus we can subsample data to calculate estimates of the operator when it derives from linear operators of the log density, such as differentiation and the identity. This follows as a linear operator of sums is a sum of linear operators, so the gradients in Equation 7 and Equation 8 decompose into a sum. The Langevin-Stein and KL operator are both linear in the log density; both support data subsampling.

## 3.2 Variational Programs

Given an operator and variational family, Algorithm 1 optimizes the corresponding operator objective. Certain operators require the density of $q$. For example, the KL operator (Equation 5) requires its log density. This potentially limits the construction of rich variational approximations for which the density of $q$ is difficult to compute.[1]

Some operators, however, do not depend on having a analytic density; the Langevin-Stein (LS) operator (Equation 3) is an example. These operators can be used with a much richer class of variational approximations, those that can be sampled from but might not have analytically tractable densities. We call such approximating families *variational programs*.

Inference with a variational program requires the family to be reparameterizable [11, 22]. (Otherwise we need to use the score function, which requires the derivative of the density.) A reparameterizable variational program consists of a parametric deterministic transformation $R$ of random noise $\epsilon$. Formally, let

$$\epsilon \sim \text{Normal}(0, 1), \quad z = R(\epsilon; \lambda). \tag{9}$$

This generates samples for $z$, is differentiable with respect to $\lambda$, and its density may be intractable. For operators that do not require the density of $q$, it can be used as a powerful variational approximation. This is in contrast to the standard Kullback-Leibler (KL) operator.

As an example, consider the following variational program for a one-dimensional random variable. Let $\lambda_i$ denote the $i$th dimension of $\lambda$ and make the corresponding definition for $\epsilon$:

$$z = (\epsilon_3 > 0)R(\epsilon_1; \lambda_1) - (\epsilon_3 \leq 0)R(\epsilon_2; \lambda_2). \tag{10}$$

When $R$ outputs positive values, this separates the parametrization of the density to the positive and negative halves of the reals; its density is generally intractable. In Section 4, we will use this distribution as a variational approximation.

Equation 9 contains many densities when the function class $R$ can approximate arbitrary continuous functions. We state it formally.

**Theorem 1.** *Consider a posterior distribution $p(z \mid x)$ with a finite number of latent variables and continuous quantile function. Assume the operator variational objective has a unique root at the posterior $p(z \mid x)$ and that $R$ can approximate continuous functions. Then there exists a sequence of parameters $\lambda_1, \lambda_2 \ldots$, in the variational program, such that the operator variational objective converges to 0, and thus q converges in distribution to $p(z \mid x)$.*

This theorem says that we can use variational programs with an appropriate $q$-independent operator to approximate continuous distributions. The proof is in Appendix D.

# 4 Empirical Study

We evaluate operator variational inference on a mixture of Gaussians, comparing different choices in the objective. We then study logistic factor analysis for images.

## 4.1 Mixture of Gaussians

Consider a one-dimensional mixture of Gaussians as the posterior of interest, $p(z) = \frac{1}{2}\text{Normal}(z; -3, 1) + \frac{1}{2}\text{Normal}(z; 3, 1)$. The posterior contains multiple modes. We seek to approximate it with three variational objectives: Kullback-Leibler (KL) with a Gaussian approximating family, Langevin-Stein (LS) with a Gaussian approximating family, and LS with a variational program.

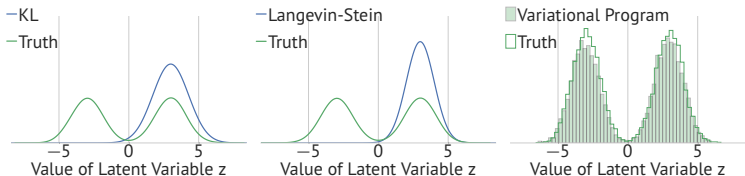

**Figure 1:** The true posterior is a mixture of two Gaussians, in green. We approximate it with a Gaussian using two operators (in blue). The density on the far right is a variational program given in Equation 10 and using the Langevin-Stein operator; it approximates the truth well. The density of the variational program is intractable. We plot a histogram of its samples and compare this to the histogram of the true posterior.

Figure 1 displays the posterior approximations. We find that the KL divergence and LS divergence choose a single mode and have slightly different variances. These operators do not produce good results because a single Gaussian is a poor approximation to the mixture. The remaining distribution in Figure 1 comes from the toy variational program described by Equation 10 with the LS operator. Because this program captures different distributions for the positive and negative half of the real line, it is able to capture the posterior.

In general, the choice of an objective balances statistical and computational properties of variational inference. We highlight one tradeoff: the LS objective admits the use of a variational program; however, the objective is more difficult to optimize than the KL.

## 4.2 Logistic Factor Analysis

Logistic factor analysis models binary vectors $x_i$ with a matrix of parameters $W$ and biases $b$,

$$z_i \sim \text{Normal}(0, 1)$$
$$x_{i,k} \sim \text{Bernoulli}(\sigma(w_k^\top z_i + b_k)),$$

where $z_i$ has fixed dimension $K$ and $\sigma$ is the sigmoid function. This model captures correlations of the entries in $x_i$ through $W$.

We apply logistic factor analysis to analyze the binarized MNIST data set [24], which contains 28x28 binary pixel images of handwritten digits. (We set the latent dimensionality to 10.) We fix the model parameters to those learned with variational expectation-maximization using the KL divergence, and focus on comparing posterior inferences.

We compare the KL operator to the LS operator and study two choices of variational models: a fully factorized Gaussian distribution and a variational program. The variational program generates samples by transforming a $K$-dimensional standard normal input with a two-layer neural network, using rectified linear activation functions and a hidden size of twice the latent dimensionality. Formally,

| Inference method | Completed data log-likelihood |
|---|---|
| Mean-field Gaussian + KL | -59.3 |
| Mean-field Gaussian + LS | -75.3 |
| Variational Program + LS | -58.9 |

**Table 1:** Benchmarks on logistic factor analysis for binarized MNIST. The same variational approximation with LS performs worse than KL on likelihood performance. The variational program with LS performs better without directly optimizing for likelihoods.

the variational program we use generates samples of $z$ as follows:

$$z_0 \sim \text{Normal}(0, I)$$
$$h_0 = \text{ReLU}(W_0^{q\top} z_0 + b_0^q)$$
$$h_1 = \text{ReLU}(W_1^{q\top} h_0 + b_1^q)$$
$$z = W_2^{q\top} h_1 + b_2^q.$$

The variational parameters are the weights $W^q$ and biases $b^q$. For $f$, we use a three-layer neural network with the same hidden size as the variational program and hyperbolic tangent activations where unit activations were bounded to have norm two. Bounding the unit norm bounds the divergence. We used the Adam optimizer [12] with learning rates $2 \times 10^{-4}$ for $f$ and $2 \times 10^{-5}$ for the variational approximation.

There is no standard for evaluating generative models and their inference algorithms [25]. Following Rezende et al. [22], we consider a missing data problem. We remove half of the pixels in the test set (at random) and reconstruct them from a fitted posterior predictive distribution. Table 1 summarizes the results on 100 test images; we report the log-likelihood of the completed image. LS with the variational program performs best. It is followed by KL and the simpler LS inference. The LS performs better than KL even though the model parameters were learned with KL.

## 5   Summary

We present operator variational objectives, a broad yet tractable class of optimization problems for approximating posterior distributions. Operator objectives are built from an operator, a family of test functions, and a distance function. We outline the connection between operator objectives and existing divergences such as the KL divergence, and develop a new variational objective using the Langevin-Stein operator. In general, operator objectives produce new ways of posing variational inference.

Given an operator objective, we develop a black box algorithm for optimizing it and show which operators allow scalable optimization through data subsampling. Further, unlike the popular evidence lower bound, not all operators explicitly depend on the approximating density. This permits flexible approximating families, called variational programs, where the distributional form is not tractable. We demonstrate this approach on a mixture model and a factor model of images.

There are several possible avenues for future directions such as developing new variational objectives, adversarially learning [5] model parameters with operators, and learning model parameters with operator variational objectives.

**Acknowledgments.**   This work is supported by NSF IIS-1247664, ONR N00014-11-1-0651, DARPA FA8750-14-2-0009, DARPA N66001-15-C-4032, Adobe, NSERC PGS-D, Porter Ogden Jacobus Fellowship, Seibel Foundation, and the Sloan Foundation. The authors would like to thank Dawen Liang, Ben Poole, Stephan Mandt, Kevin Murphy, Christian Naesseth, and the anonymous reviews for their helpful feedback and comments.

## Footnotes

[1]It is possible to construct rich approximating families with $\text{KL}(q||p)$, but this requires the introduction of an auxiliary distribution [15].

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
