[Supplementary Material]

# Supplement for Operator Variational Inference

**Rajesh Ranganath**
Princeton University

**Jaan Altosaar**
Princeton University

**Dustin Tran**
Columbia University

**David M. Blei**
Columbia University

## A  Technical Conditions for Langevin-Stein Operators

Here we establish the conditions needed on the function class $\mathcal{F}$ or the posterior distribution short-handed $p$ for the operators to have expectation zero for all $f \in \mathcal{F}$. W derive properties using integration by parts for supports that are bounded open sets. Then we extend the result to unbounded supports using limits. We start with the Langevin-Stein operator. Let $S$ be the set over which we integrate and let $B$ be its boundary. Let $v$ be the unit normal to the surface $B$, and $v_i$ be the $i$th component of the surface normal (which is $d$ dimensional). Then we have that

$$
\int_S p(O_{\mathrm{LS}}^p f) dS = \int_S p\nabla_z \log p^\top f + p\nabla^\top f dS
$$

$$
= \sum_{i=1}^d \int_S \frac{\partial}{\partial z_i}[p] f_i + p\frac{\partial}{\partial z_i}[f_i] dS
$$

$$
= \sum_{i=1}^d \int_S \frac{\partial}{\partial z_i}[p] f_i dS + \int_B f_i p v_i dB - \int_S \frac{\partial}{\partial z_i}[p] f_i dS
$$

$$
= \int_B v^\top f p dB.
$$

A sufficient condition for this expectation to be zero is that either $p$ goes to zero at its boundary or that the vector field $f$ is zero at the boundary.

For unbounded sets, the result can be written as a limit for a sequence of increasing sets $S_n \to S$ and a set of boundaries $B_n \to B$ using the dominated convergence theorem [2]. To use dominated convergence, we establish absolute integrability. Sufficient conditions for absolute integrability of the Langevin-Stein operator are for the gradient of $\log p$ to be bounded and the vector field $f$ and its derivatives to be bounded. Via dominated convergence, we get that $\lim_n \int_{B_n} v^\top f p dB = 0$ for the Langevin-Stein operator to have expectation zero.

## B  Characterizing the zeros of the Langevin-Stein Operators

We provide analysis on how to characterize the equivalence class of distributions defined as $(O^{p,q} f)(z) = 0$. One general condition for equality in distribution comes from equality in probability on all Borel sets. We can build functions that have expectation zero with respect to the posterior that test this equality. Formally, for any Borel set $A$ with $\delta_A$ being the indicator, these functions on $A$ have the form:

$$
\delta_A(z) - \int_A p(\mathbf{y}) d\mathbf{y}
$$

We show that if the Langevin-Stein operator satisfies $\mathcal{L}(q; O_{\mathrm{LS}}^p, \mathcal{F}) = 0$, then $q$ is equivalent to $p$ in distribution. We do this by showing the above functions are in the span of $O_{\mathrm{LS}}^p$. Expanding the

Langevin-Stein operator we have

$$(O_{\text{LS}}^p f) = p^{-1} \nabla_z p^\top f + \nabla^\top f = p^{-1} \sum_{i=1}^{d} \frac{\partial f_i p}{\partial z_i}.$$

Setting this equal to the desired function above yields the differential equation

$$\delta_A(z) - \int_A p(y)dy = p^{-1}(z) \sum_{i=1}^{d} \frac{\partial f_i p}{\partial z_i}(z).$$

To solve this, set $f_i = 0$ for all but $i = 1$. This yields

$$\delta_A(z) - \int_A p(y)dy = p^{-1}(z) \frac{\partial f_1 p}{\partial z_1}(z),$$

which is an ordinary differential equation with solution for $f_1$

$$f_1^A(z) = \frac{1}{p(z)} \int_{-\infty}^{z_1} p(a, z_{2\ldots d}) \left( \delta_A(a, z_{2\ldots d}) - \int_A p(y)d\,y \right) da.$$

This function is differentiable with respect to $z_1$, so this gives the desired result. Plugging the function back into the operator variational objective gives

$$\mathbb{E}_q \left[ \delta_A(z) - \int_A p(y)d\,y \right] = 0 \iff \mathbb{E}_q[\delta_A(z)] = \mathbb{E}_p[\delta_A(z)],$$

for all Borel measurable $A$. This implies the induced distance captures total variation.

## C   Operators for Discrete Variables

Some operators based on Stein's method are applicable only for latent variables in a continuous space. There are Stein operators that work with discrete variables [1, 4]. We present one amenable to operator variational objectives based on a discrete analogue to the Langevin-Stein operator developed in [4]. For simplicity, consider a one-dimensional discrete posterior with support $\{0, \ldots, c\}$. Let $f$ be a function such that $f(0) = 0$, then an operator can be defined as

$$(O_{\text{DISCRETE}}^p f)(z) = \frac{f(z+1)p(z+1, x) - f(z)p(z, x)}{p(z, x)}.$$

Since the expectation of this operator with respect to the posterior $p(z \mid x)$ is a telescoping sum with both endpoints 0, it has expectation zero.

This relates to the Langevin-Stein operator in the following. The Langevin-Stein operator in one dimension can be written as

$$(O_{\text{LS}}^p f) = \frac{\frac{d}{dz}[fp]}{p}.$$

This operator is the discrete analogue as the differential is replaced by a discrete difference. We can extend this operator to multiple dimensions by an ordered indexing. For example, binary numbers of length $n$ would work for $n$ binary latent variables.

## D   Proof of Universal Representations

Consider the optimal form of $R$ such that transformations of standard normal draws are equal in distribution to exact draws from the posterior. This means

$$R(\epsilon; \lambda) = P^{-1}(\Phi(\epsilon)),$$

where $\Phi(\epsilon)$ squashes the draw from a standard normal such that it is equal in distribution to a uniform random variable. The posterior's inverse cumulative distribution function $P^{-1}$ is applied to the

uniform draws. The transformed samples are now equivalent to exact samples from the posterior. For a rich-enough parameterization of $R$, we may hope to sufficiently approximate this function.

Indeed, as in the universal approximation theorem of Tran et al. [5] there exists a sequence of parameters $\{\lambda_1, \lambda_2, \ldots\}$ such that the operator variational objective goes to zero, but the function class is no longer limited to local interpolation. Universal approximators like neural networks [3] also work. Further, under the assumption that $p$ is the unique root and by satisfying the conditions described in Section B for equality in distribution, this implies that the variational program given by drawing $\epsilon \sim \mathcal{N}(\mathbf{0}, \mathbf{I})$ and applying $R(\epsilon)$ converges in distribution to $p(z \,|\, x)$.