[Reviews · NeurIPS 2016]

Reviewer 1

Summary

The authors consider generalizations of standard variational inference objective function based on operator methods. Within this generalized class of objective functions, new objectives with desirable properties ( compatible with mini-batches and "variational programs" ) can be obtained. The standard KL objective is a particular case of the proposed generalizations. The authors propose a few experiments on density estimation on data generated by a Gaussian mixture model and on MNIST.

Qualitative Assessment

The paper is overall very well organized and the derivations are sound. The experimental results are somewhat unsatisfying. See details below: 1) The "variational program" in equation (3) defines directly a mixture of two Gaussians, which is very close to how the data was generated for the Gaussian Mixture experiments. The result would be more powerful if the authors had used a less specific functional form, such as an MLP. 2) On 260, what is in "... T as a linear … " ? does it refer to t(a) in equation (1) being the identity? 3) It would help if the authors add more details about the parametrization used for f in the supplementary materials. 4) From lines 289-291 it seems that the likelihoods in Table 1 were computed conditioned on half of the pixels instead of being computed unconditionally. It is not clear to me the motivation for this. Please clarify. Minor: 1) Isn't there a sup_f missing in the objective defined in Lines 182-183? 2) Fix brackets in equation (9)

Confidence in this Review

2-Confident (read it all; understood it all reasonably well)


Reviewer 2

Summary

This paper proposes an operator view of variational Bayesian inference which gives an unified view of VB w.r.t. different types of divergence functions as well as conditions for tractable optimization. The variatioal objective is defined on an operator and the authors derive different algorithms by adopting different operators though the optimization procedure is unified as operator variational Bayes which is a kind of black box VB. Experiments were conducted on two very simple datasets and the proposed operator VB was compared with auto-encoder VB and showed an advantage of their method.

Qualitative Assessment

Novelty: Although there exists a lot of efforts to extend traditional KL-divergence VB to wider class of divergence functions, I haven't never seen this idea of operator view of VB and it is technically interesting. I think it is a nice contribution that this view unifies many recent ideas around VB. Presentation: Presentation is good and can be improved by clarifying a few notions. - I'm a bit confused by test functions f and \mathcal{F}. It is just mentioned as "parametric functions" but there is no concrete definition or examples to understand what is f actually. For example, in line 112, there is no f in the right hand side of variational objective of KL divergence. In the experiments, no explanation of how to select \mathcal{F} (or even what is \mathcal{F} in this case). - Eq. (2) and (3) are difficult to understand due to lack of definition of R and \lambda. Potential Impact or usefulness: I gave a low score because of a few concerns. 1. Experiments were conducted only on very simple datasets and mainly visual evaluation. I'm not sure if this method practically improves performance of learning latent variable models. The authors claim that it is hard to evaluate experiment 2 because different methods uses different optimization criteria. This is true. However, they can be still compared based on for example classification accuracy (i..e using learned latent variables as features and learn classifiers). 2. I guessed this paper is motivated to gain robustness in learning deep neural network. It is better to have analysis (and empirical evaluations) on computational efficiency since computational efficiency as well as tractability is one of the most important aspect of recent deep neural network learning. Questions: - the role of distance function t is unclear. How does it affect the result? The authors provides only two simple examples (squared and identity). Any other practically or theoretically important t? -

Confidence in this Review

2-Confident (read it all; understood it all reasonably well)


Reviewer 3

Summary

The paper proposes a generalization of the variational inference framework, which contains the standard KL divergence objective as a special case. Based on the new framework, the authors proposed new objectives, and showed some illustrative examples. After rebuttal: I understand what f and F are, and the whole story. In my opinion, the paper would be much easier to read if it starts from explaining Stein's method, and then establish the variational inference framework. Showing KL as the simplest example never helps and only confusing. Anyway, I think this is an interesting paper.

Qualitative Assessment

I understand Sections 1 and 2, but not Sections 3 and 4. The main problem is that I cannot get what f and F mean, and why supremum over F is taken in (1). It is sometimes called a test function, but I don't know what the authors call a test function. It's clear that (1) corresponds to the KL divergence with the operator defined in the end of Section 2.2. But this example doesn't help readers understand the role of f and F. Optimally, I'd like the authors to give the second simplest example, where the operator depends on f, F has multiple instances, but the operator is still simple. Otherwise (or if the stein divergence is the second simplest example), please explain more about Eq.(4). What are f and F? Are they related to the posterior p(z|x) or the approximate distribution q(z)? Can you define what is the test function? More explaintion on "variational program" is also preferable. Does it define q as a generative model, from which one can get samples but its density is unknown? Perhaps, more separate discussion would help on how the authors adopt different divergence and how the authors broaden the class of possible q. Since I don't understand the main part (my confidence level is below 1), my score is temporary and likely to be be changed, depending on author's rebuttal and the discussion between reviewers. I hope I can really understand the paper with author's response, since the paper seems to contain a signifcant contribution. Minor comments: - In the equation below (5), sup_y is missing?

Confidence in this Review

2-Confident (read it all; understood it all reasonably well)


Reviewer 4

Summary

This paper proposed operator variational inference where the objective is a broad yet tractable set of possible optimization for approximating the posterior distribution. The proposed framework provided new objectives for variational inference via Stein’s method.

Qualitative Assessment

The generalization of the variational inference by using an operator is novel. This extension provides a novel variational inference via Stein's method, which means that the generalization is not only interesting but also useful.

Confidence in this Review

2-Confident (read it all; understood it all reasonably well)


Reviewer 5

Summary

This paper proposes a class of variational inference methods by minimizing the worse case value among all these functions after applying an operator. Named operator variational inference, this method extends variational inference to q distributions with intractable density. Stein's method is applied to construct operators and the paper lists two examples. Black-box methods is also developed.

Qualitative Assessment

Traditional variational methods requires tractable q distributions and this paper tries to relax this requirement: we only need to be able to sample from q. Given that now the trend of VI is to introduce MC methods and use deep neural networks to specify approximations, this work is very timely and I haven't seen any previous work on this topic. The authors presented the main idea in a clear way, although the new operator examples are less intuitive to me. Experiment results looks promising. A few questions and comments: 1. What's your definition of "being variational"? As far as I understand, I will call a method which uses the approximate posterior to help approximate the hyper-parameter optimization objective as "variational". When you construct the operator using f-divergence that's no problem. But for more general case, e.g. Stein's method you discussed, I don't see how to optimize the hyper-parameters for those objectives. There's no doubt that the proposed method is an approximate inference algorithm: I'm just a bit unsure about the name. 2. For the black-box method, I think you need to assume the operator to be independent or linear to log q? 3. When using neural networks for both f and q the minimax problem can be very unstable if you directly apply gradient descent. This is the main problem for generative adversarial networks. Did you have this issue in your experiments? 4. In the DLGM experiment, did you fix a pre-trained generative model to evaluate the variational approximation? This links to question 1 that I'm not sure how to learn hyper-parameters with say Stein operator. Also how did you compute the likelihood values?

Confidence in this Review

2-Confident (read it all; understood it all reasonably well)


Reviewer 6

Summary

The paper introduces Operator Variational Inference, that defines novel variational objectives by using operators and extends standard variational approaches based on the minimization of the KL divergence. Operator Variational Inference is efficient even without the standard mean field assumption, and it allows to approximate the posterior with a broad class of flexible distributions. When using the reparametrization gradient, the introduced variational objectives do not require the variational distribution q to be analytically tractable, and one can only learn how to sample from it with a Variational Program.

Qualitative Assessment

There has been a lot of interest recently in defining variational objectives that improve upon the standard variational bound, as minimizing KL[q||p] leads to variational distributions that severely underestimate the variance of the posterior. By using operators, this paper introduces a novel way to form variational objectives that allow to fit a more flexible class of variational approximations, either by optimizing the parameters of q or by learning how to sample from it (variational programs). I find this approach very interesting, and a promising research direction. The presentation of the paper, especially in the theoretical part, is sometimes confusing and difficult to understand: - I found section G/H/I/J of the supplementary material necessary to understand more intuitively Operator Variational Inference. They should be put in the main paper, or at least referenced more explicitly and using section names. - In lines 144-161 you only introduce in detail the type of distributions obtained with the reparametrization, but in section 4 you also present the score function gradient. This is confusing, especially as it is only said in the supplementary material that in this case it is needed an analytically tractable variational distribution, as needed in equation 8. The mixture of Gaussian experiment is useful to understand the behaviour of the different variational objectives, but it looks strange that when using the KL objective the variational approximation does not pick only one of the modes. The MNIST experiment is not enough to assess how the different methods perform. You may consider using more that 10 units (e.g. 100) and reporting how much time it takes for the different methods to converge (with "Stein" you also need to optimize the test function). More importantly, I would have liked to see results also on different data sets. Finally, an interesting related work is "Variational Inference with Renyi Divergence", Y Li, RE Turner - arXiv preprint arXiv:1602.02311, 2016

Confidence in this Review

2-Confident (read it all; understood it all reasonably well)